# Effect of eucalyptus wood-based compost application rates on soil chemical properties in semi-organic avocado plantations, Limpopo province, South Africa

P. M. Mohale[1], A. Manyevere[1,2]*, C. Parwada[2,3], M. G. Zerizghy[1]

1 Faculty of Science and Agriculture, School of Agriculture and environmental Science, University of Limpopo, Polokwane, Limpopo, South Africa, 2 Faculty of Science and Agriculture, Department of Agronomy, University of Fort Hare, Alice, South Africa, 3 Faculty of Agricultural Sciences, Department of Agricultural Management, Zimbabwe Open University, Hwange, Zimbabwe

* amanyevere@ufh.ac.za

**Data Availability Statement:** All relevant data are within the paper.

## Abstract

Mixing different types of organic matters to form a compound compost can be useful in both short- and long-term improvement of soil chemical properties. However, effects of such composts on soil chemical properties are unknown. A 3-year field study was done to determine the effects of eucalyptus wood-based compost on selected chemical properties of soils at Mooketsi and Politsi avocado orchards, Limpopo Province, South Africa. The study was laid as a repeated measures design with 4 compost treatments at 0, 5, 10, and 15 t ha$^{-1}$ year$^{-1}$ with three replicates. Soil pH, EC, organic carbon, active carbon, soil $NO_3$-N, $NH_4$-N, PMN, P K, Ca, Mg, Na Mn, Cu, Zn, and Fe were measured annually at the two orchards after applying compost at the different rates. The eucalyptus wood-based compost significantly ($p < 0.05$) altered the composition of the measured chemical properties at both orchards. The values of the soil chemical properties increased cumulatively with compost quantity and time. This trend was consistent at both orchards. Significantly ($p < 0.05$) highest and lowest values of the soil chemical properties were recorded at 15t ha$^{-1}$ and 0t ha$^{-1}$ in 2018 respectively but with no significant ($p > 0.05$) differences between 10t ha$^{-1}$ and 15t ha$^{-1}$ compost application rates the entire study period. Eucalyptus wood-based compost raised values of the soil chemical properties at $\geq$10t ha$^{-1}$ application rates. It is recommended to apply the eucalyptus wood-based compost $\geq$10t ha$^{-1}$ at both orchards, nevertheless, other factors governing organic matter decomposition such as soil temperature were not measured hence further studies are necessary.

## Introduction

Maintenance of a constantly suitable soil nutrient supply is important in achieving uniformly large avocado fruits [1]. Nevertheless, the maintenance of the favorable soil chemical status in avocado orchards is still challenging especially under an inorganic fertilizer regime. Compost

**Funding:** The study did not receive any specific grant although the first author received some financial support from the NRF and the ZZ2 Bertie van Zyl (Pty) Ltd, South Africa. The funders had no role in study design, data collection and analysis, decision to publish, or preparation of the manuscript.

**Competing interests:** The authors have declared that no competing interests exist.

could be a significant reservoir of nutrients that can supply nutrients to the plants [2, 3]. The compost has several advantages when applied to the soil and these include improved soil physical, chemical, and biological properties and provision of important nutrients for plant uptake [2]. The effect of compost on soil chemical properties is expounded to the increased organic carbon content and cation exchange capacity (CEC) of the soil [4]. Compost is a source of soil organic matter (SOM), which influences the availability of the essential elements through biogeochemical processes like the carbon, nitrogen, phosphorus, and sulfur cycles [5]. The compost is usually seen as a decent source of nitrogen (N), phosphorous (P), potassium (K), micronutrients, and humic substances essential for plant growth [6]. Soil nutrients like N, P, K, Ca, Mg, Zn, Cu, B, and Fe are important for the production of fruit trees like avocado [15]. These nutrients should be readily available to the trees because they are required in the various physiological functioning of the plant ranging from germination up to fruit ripening [7].

The compost is usually applied in combination with chemical fertilizers so as to achieve the maximum benefits [4]. Using compost along with inorganic fertilizers is pricey and poses a risk of nutrients leaching [6, 8]. However, there is a possible influence of nutrients once they are insufficient or in excess range to a tree [7, 9]. Excessive N application resulted in a luxurious uptake of the nutrient which leads to rank growth at the expense of flower and fruit production, increased fragility of the stems and branches, and cause delayed fruit set in an avocado [5]. Conversely, inadequate N supply cause stunted growth and reduced chlorophyll content in the leaves of avocado trees [9–11]. This reduces the rate of photosynthesis and resulted in reduced crop productivity.

Limited ($< 2.1$ kg ha$^{-1}$) supply of Ca to the avocado causes roots to collapse and disintegrate whereas deficiency ($< 0.04$ kg ha$^{-1}$) of Zn will reduce leaf and shoots growth, yield, and also lowers tree health [9]. Excess ($> 0.8$ kg ha$^{-1}$) supply of Na to the avocado tree results in the drying out of both young and mature branches [11]. Management of the availability of the nutrients is very important especially the micronutrients as the avocado trees have low mineral nutrient requirements [5, 9, 10, 12]. This assumption is premised on the fact few mineral deficiencies mainly of N ($< 11.3$ kg ha$^{-1}$), Zn ($< 0.04$ kg ha$^{-1}$), and Fe ($< 0.09$ kg ha$^{-1}$) have been found in commercial orchards. The application of fertilizers using either organic or inorganic sources is one of the foremost common cultural management practices in avocado production [5].

Commercial agriculture has been and still relies on the use of inorganic fertilizers for increasing crop yield [13]. However, the inorganic fertilizers do contribute substantially to environmental degradation and promote high loss of nutrients through leaching [14, 15]. The high losses of nutrients are caused by the poor health of the soil system leading to a reduction in the cation exchange capacity of the soil [5]. Compost application can modify the soil properties and enhance the slow release of nutrients over time, increasing soil cation exchange capacity and improving root proliferation [4, 7]. The use of compost in improving soil nutrient holding capacity and rate of release has been investigated for several years in several crops [13]. However, information on the composition of the compost and application rates in different soils is still hazy [9].

The ZZ2 Bertie van Zyl (Pty) Ltd is a reputable avocado producer which is both environmentally and consumer health conscious hence aims to use sustainable organic farming models that reduce the use of inorganic fertilizers [16]. The soils at the two major ZZ2 Bertie van Zyl (Pty) Ltd orchards, Mooketsi and Politsi are Nitisols and Ferralsols respectively [16] and are acidic soils. The soil types, particularly the Ferralsols is high in iron and aluminum oxides resulting crystallization and high P-adsorption capacities [7]. This suggests that the soils are unfertile so the organic fertilizers are ideal in that they promote a continuously release plant nutrients thereby achieving uniform growth rates in plants. The organic

fertilizers promote the formation of soil aggregates that improve soil friability, which facilitate plant growth, prevent waterlogging and root asphyxia [9]. Sustainable health of the soil relies on carbon-rich amendments that promote the biological processes which are fundamental for healthy soil. The ZZ2 Bertie van Zyl (Pty) Ltd is currently using compost in combination with inorganic fertilizers in the avocado orchards to improve soil fertility and enhancing fruit yields. It is proposing to intensify the organic farming practices on the orchards by adopting a mixed source of organic compost. Unfortunately, there is limited information on the effect of applying the compost alone in maintaining and improving the soil chemical properties in avocado orchards. Therefore, the objectives of this study were to determine the effects of eucalyptus-based compost on selected chemical properties of soils under two 'Hass' avocado plantations. It had been hypothesized that eucalyptus compost application rate and time-post application are important in modifying the soil chemical properties and nutrient release of the soils.

## Materials and methods

The study was conducted at two avocado orchards in the Limpopo province, South Africa. The orchards are Mooketsi (23 038'53.563" S, 30 03'18.964" E and 751 m altitude) and Politsi orchards (23 045'56.405" S, 30 06'42.264" E and 805 m altitude). Both orchards are in the subtropics, with a mean annual rainfall of 881 mm and 550 mm, as well as an average temperature of 20.30C and 20.50C at Politsi and Mooketsi respectively. Soils at the Mooketsi and Politsi can be classified as Nitisols and Ferralsols respectively [17]. The avocado variety was established in 2007 by grafting 'Hass' scion on a 'Dusa' rootstock. Avocado trees were spaced at 10 m × 5 m and each tree had 2 micro-sprinklers with 250 cm -380 cm wetting diameter on the soil surface. Irrigation scheduling determined by a tensiometer that was installed in each orchard. The irrigation water had an EC of < 0.5 dS m$^{-1}$ and the soil water was maintained between 30 and 40 kpa and >60% water holding capacity for the optimum growth of the avocado trees. Fertilizer was applied according to the soil test results with 100 kg N ha$^{-1}$ year$^{-1}$ and 55 kg K ha$^{-1}$ year$^{-1}$, and 170 kg N ha$^{-1}$ year$^{-1}$ and 53 kg K ha$^{-1}$ year$^{-1}$ were split applied at Mooketsi and Politsi respectively.

### Design and experimental treatments

The experiments within the orchards were laid in a repeated measures design with four compost treatments that were 0, 5, 10, and 15 t ha$^{-1}$ year$^{-1}$ with three replicates. The 10 t ha$^{-1}$ was used as the median compost application rate in the study based on recommendations for remediation and soil nutrition at the Mooketsi and Politsi orchards. An experimental plot comprised of 12 rows with 10 trees in each row. Each row represented a treatment replicate. Compost was applied under the tree canopy on the soil surface employing a Kuhn knight compost spreader.

**Eucalyptus compost preparation, sampling and analysis.** The compost was prepared through controlled decomposition of organic material under aerobic conditions that allowed the event of thermophilic temperature. Compost piles measuring 18 m × 2 m × 1.5 m were designed on-site from locally available eucalyptus wood chips (40%) and sawdust (20%), manure (20%), and cattle manure (20%) stock feed [18]. The piles were mechanically turned after every 10 days to allow air circulation and homogenizing the compost. The moisture content in the compost pile was adjusted watering to maintain optimum composting conditions (approximately 40–45% moisture). The maturity of the compost was evaluated on a monthly basis throughout the process by monitoring the temperature with a long probe thermometer to a depth of 0.60 m. At maturity, the compost was tested for toxicity to plants by planting

lettuce and onion seeds. The seeds were selected for phytotoxicity bioassay because they have a good response to toxic materials and rapid germination [18]. Compost samples were collected from four different windrows at three depths using a shovel and tested for phytotoxicity as explained by [18].

## Data collection

Soil samples were collected from each tree in a row before compost was applied and for 3 consecutive years after the compost was applied in 2016, 2017, and 2018. Soil samples were collected at peripheral, intermediate, and proximal areas to the trunk of the tree at a depth of 30 cm using a graduated soil auger. Soil samples from the 10 trees were mixed thoroughly to obtain a composite sample. The composite samples were air-dried and passed through a 2-mm sieve. Soil pH and electrical conductivities (ECs) were measured in a soil-water suspension (ratio of 1: 5) using a TPS meter as described by [19]. Organic carbon was determined by the modification of wet acid digestion of the Walkley-Black method [20]. Total nitrogen (N) determination, P, and exchangeable ammonium and nitrate, nitrite, Cu, Zn, Fe, and Mn in the soil were analyzed as described by [21].

## Data analysis

Analysis of variance (ANOVA) to determine the effects of the eucalyptus-based compost application rates on the changes of soil chemical properties were done using the Statistix software version 10.0. Fisher's least significant difference (LSD) test was accustomed separate treatment means and soils chemical norms at a 5% confidence level. Soil nutrient level results obtained from the Politsi and Mooketsi orchards after applying compost at different rates were categorized using critical soil nutrients limits for avocado production adopted by ARC-ITSC Nelspruit, African country (Table 1).

## Ethical considerations

There were no human or animal subjects in this study and informed consent is not applicable.

**Table 1. Important soil nutrients categories for avocado production as recommended by the ARC-ITSC Nelspruit, South Africa.**

| Nutrient | Units | Low | Sufficient | High | Reference |
|---|---|---|---|---|---|
| pH (H$_2$O) | - | <5.5 | 5.5–6.5 | <6.5 | Schaffer et al., 2013 |
| EC | dS/m | - | - | >4 | Crowley, 2008 |
| N | mg kg$^{-1}$ | <25 | - | - | Joubert, 2016 |
| P (Bray 1) | mg kg$^{-1}$ | <18 | 18–60 | >60 | Schaffer et al., 2013 |
| K | mg kg$^{-1}$ | <70 | 70–250 | >250 | Schaffer et al., 2013 |
| Ca | mg kg$^{-1}$ | <500 | 500–2000 | >2000 | Joubert, 2016 |
| Mg | mg kg$^{-1}$ | <200 | 200–400 | >400 | Schaffer et al., 2013 |
| Na | mg kg$^{-1}$ | - | <20 | >20 | Joubert, 2016; Abercrombie, 2009 |
| Mn | mg kg$^{-1}$ | <6 | 6–40 | >40 | Joubert, 2016 |
| Cu | mg kg$^{-1}$ | <3 | 3–10 | >10 | Joubert, 2016 |
| Zn | mg kg$^{-1}$ | <5 | 5–20 | >20 | Joubert, 2016 |
| Fe | mg kg$^{-1}$ | <4 | 4–20 | >20 | Joubert, 2016 |
| Organic Carbon | % | - | >2 | - | Schaffer et al., 2013 |

Source, [22]

## Results

The soil texture was 45% clay, 19% silt, and 36% sand and 24% clay, 5% silt, and 71% sand at Politsi and Mooketsi respectively. Soil pH was acid (below pH 7) to neutral for both Mooketsi and Politsi (Table 2). The soils at both orchards have sufficient micronutrients contents, however with low content ($<18$ mg kg$^{-1}$) of P, and NO$_3$-N ($<25$ mg kg$^{-1}$) (Tables 1 & 2). Soils at the Mooketsi and Politsi have EC$<1$ dS m$^{-1}$ therefore are classified as non-saline soils. The concentrations of NO$_3$-N and NH$_4$-N at the Mooketsi and Polistsi were below the moderate (25–50 mg-N/kg) soil nitrogen supply levels (Table 2).

The moisture content and temperature of the compost were within the range of matured compost. The compost had a C: N ratio of 23.6:1 and was non-saline because of the EC $<$ 1dS m$^{-1}$ (Table 3). Onion and lettuce seed germination were also $>$ 70%, which indicates that the compost was not toxic to plants (Table 3).

There were significant ($p<0.05$) interactions between the eucalyptus-wood base compost application rates (R) $\times$ time (Y) on the soil chemical properties at both the Mooketsi and Politsi orchards (Table 4).

### Mooketsi orchard

**Soil pH, EC, organic carbon and active carbon.**   The soil pH was acidic ($<7$) throughout the 3 years under 0 t ha$^{-1}$ application rates but became alkaline ($>7$) in 2017 and 2018 under $>0$ t ha$^{-1}$ compost application rates (Table 5). The EC was lowest (0.2 dS m$^{-1}$) and highest (1.2 dS m$^{-1}$) in 2018 under 0 t ha$^{-1}$ and 15 t ha$^{-1}$ compost application rates respectively. The

**Table 2. Selected chemical and physical soil properties at the Mooketsi and Politsi avocado orchards before application of the compost.** (n = 4).

| Chemical properties | Units | Mooketsi | Politsi |
|---|---|---|---|
| pH (H$_2$O) | - | 6.6 | 6.7 |
| EC | dS m$^{-1}$ | 0.7 | 0.8 |
| NO$_3$-N | mg kg$^{-1}$ | 7.3 | 11.8 |
| NH$_4$-N | mg kg$^{-1}$ | 8.0 | 5.3 |
| Organic carbon | mg kg$^{-1}$ | 2.0 | 2.0 |
| Active carbon | mg kg$^{-1}$ | 288 | 489 |
| Potentially mineralized nitrogen (PMN) | µgN g$^{-1}$ DM$^{-1}$ week$^{-1}$ | 2.7 | 3.0 |
| **Plant available nutrients** | | | |
| P | mg kg$^{-1}$ | 10.5 | 10.3 |
| K | mg kg$^{-1}$ | 217 | 175 |
| Ca | mg kg$^{-1}$ | 1883 | 984 |
| Mg | mg kg$^{-1}$ | 380 | 124 |
| Na | mg kg$^{-1}$ | 61.8 | 39.3 |
| Fe | mg kg$^{-1}$ | 6.0 | 5.5 |
| Mn | mg kg$^{-1}$ | 21.2 | 15.2 |
| Cu | mg kg$^{-1}$ | 4.6 | 5.4 |
| Zn | mg kg$^{-1}$ | 17.6 | 18.8 |
| **Physical properties** | | | |
| Bulk density | g ml$^{-1}$ | 1.18 | 1.00 |
| Clay | % | 45 | 24 |
| Loam | % | 19 | 5 |
| Sand | % | 36 | 71 |

**Table 3. Phytotocity test and chemical properties of the eucalyptus-based compost.** (n = 4).

| Property | Unit | Value |
|---|---|---|
| **Seed germination (phytotoxicity)** | | |
| Onion | % | 94.00 |
| Lettuce | % | 75.00 |
| **Chemical properties** | | |
| pH | - | 6.91 |
| EC | dS m$^{-1}$ | 0.82 |
| Organic Carbon | g kg$^{-1}$ | 496.0 |
| Active carbon | g kg$^{-1}$ | 569.0 |
| Potentially mineralization nitrogen (PMN) | g kg$^{-1}$ | 3.4 |
| **Total nutrients** | | |
| NO$_3$-N | g kg$^{-1}$ | 21.0 |
| NH$_4$-N | g kg$^{-1}$ | 18.4 |
| PO$_4$ | g kg$^{-1}$ | 16.7 |
| K | g kg$^{-1}$ | 69.1 |
| Ca | g kg$^{-1}$ | 0.45 |
| Mg | g kg$^{-1}$ | 0.07 |
| Na | g kg$^{-1}$ | 0.91 |
| Fe | g kg$^{-1}$ | 0.02 |
| Mn | g kg$^{-1}$ | 0.1 |
| Cu | g kg$^{-1}$ | 0.01 |
| Zn | g kg$^{-1}$ | 0.04 |

N.B. The amounts of nutrients only reflect the total amounts in compost and not plant-available nutrients

active C had an average annual increase of 5.4% under 10 t ha$^{-1}$ and 13.4% under the 15 t ha$^{-1}$ compost application rate (Table 5).

**Soil NO$_3$-N, NH$_4$-N, PMN and P.** The application of the eucalyptus wood-based compost resulted in a significant increase of the soil NO$_3$-N, NH$_4$-N, PMN, and P during the 3-year

**Table 4. ANOVA for pH, EC and organic carbon, and soil nutrient elements following a 3 year application of the eucalyptus–wood based compost at the Mooketsi and Pilisti orchards.**

| **Mooketsi orchard** | | | | | | | | | | | | | | | | |
|---|---|---|---|---|---|---|---|---|---|---|---|---|---|---|---|---|
| *Source of variation* | | pH | EC | OC | NO$_3$-N | NH$_4$-H | PMN | P | K | Ca | Mg | Na | Mn | Cu | Zn | Fe |
| Compost rate (R) | $F_{prob.}$ | 2.15 | 5.10 | 2.47 | 1.71 | 14.11 | 1.26 | 1.88 | 1.15 | 4.10 | 1.47 | 1.21 | 4.12 | 1.56 | 1.78 | 1.18 |
| | P | 0.01 | 0.02 | 0.03 | <0.01 | <0.01 | 0.02 | 0.01 | 0.01 | 0.02 | 0.03 | <0.01 | <0.01 | 0.01 | 0.01 | 0.01 |
| Time (Y) | $F_{prob.}$ | 1.92 | 2.15 | 1.50 | 1.30 | 21.80 | 16.2 | 5.67 | 1.32 | 2.55 | 0.90 | 0.90 | 3.40 | 11.2 | 25.06 | 15.67 |
| | P | <0.01 | <0.01 | <0.01 | 0.05 | <0.01 | <0.01 | <0.01 | <0.01 | <0.01 | <0.01 | 0.03 | <0.01 | <0.01 | <0.01 | <0.01 |
| R x Y | $F_{prob.}$ | 3.21 | 2.27 | 3.47 | 1.92 | 4.91 | 1.44 | 2.71 | 1.21 | 2.97 | 1.17 | 2.12 | 2.82 | 1.54 | 2.71 | 1.01 |
| | P | 0.04 | <0.01 | <0.01 | 0.03 | <0.01 | 0.01 | 0.02 | 0.04 | <0.01 | <0.01 | 0.03 | <0.01 | 0.01 | 0.02 | 0.01 |
| **Polisti orchard** | | | | | | | | | | | | | | | | |
| Compost rate (R) | $F_{prob.}$ | 3.19 | 1.10 | 2.10 | 1.20 | 16.20 | 18.20 | 1.79 | 1.19 | 4.14 | 1.38 | 1.23 | 5.25 | 12.20 | 1.90 | 0.98 |
| | P | <0.01 | <0.01 | <0.01 | <0.01 | <0.01 | <0.01 | <0.01 | <0.01 | <0.01 | <0.01 | <0.01 | <0.01 | <0.01 | <0.01 | <0.01 |
| Time (Y) | $F_{prob.}$ | 2.76 | 2.76 | 1.79 | 2.00 | 6.15 | 1.34 | 1.47 | 1.76 | 2.66 | 0.89 | 1.00 | 2.15 | 1.32 | 22.67 | 14.70 |
| | P | 0.01 | 0.01 | <0.01 | <0.01 | <0.01 | 0.01 | 0.04 | 0.01 | 0.01 | <0.01 | <0.01 | <0.01 | 0.01 | 0.04 | 0.02 |
| R x Y | $F_{prob.}$ | 2.34 | 2.34 | 3.76 | 3.18 | 11.69 | 9.35 | 49.29 | 1.34 | 2.94 | 1.06 | 2.18 | 2.69 | 6.35 | 2.29 | 19.25 |
| | P | <0.01 | <0.01 | <0.01 | 0.01 | <0.01 | <0.01 | <0.01 | <0.01 | <0.01 | <0.01 | 0.01 | <0.01 | <0.01 | <0.01 | <0.01 |

**Table 5. Soil pH, EC, organic carbon and active carbon as influenced by compost application rates over three years (2016–2018) at the Mooketsi orchard.**

| Treatment | pH (H₂O) | | | EC (dS m⁻¹) | | | Organic C (mg kg⁻¹) | | | Active C (mg kg⁻¹) | | |
|---|---|---|---|---|---|---|---|---|---|---|---|---|
| | 2016 | 2017 | 2018 | 2016 | 2017 | 2018 | 2016 | 2017 | 2018 | 2016 | 2017 | 2018 |
| 0 t ha⁻¹ | 6.6b | 5.9b | 5.6c | 0.3d | 0.3d | 0.2d | 2.1c | 1.1d | 0.8d | 138.6c | 93.3c | 72.0c |
| 5 t ha⁻¹ | 6.7ab | 7.3a | 7.5b | 0.6c | 0.5c | 0.6c | 50.3b | 60.5c | 70.0c | 347.6b | 470.0b | 495.6b |
| 10 t ha⁻¹ | 6.7ab | 7.2a | 7.8ab | 0.8ab | 0.8ab | 0.8ab | 82.5a | 88.5b | 92.5b | 450.6a | 506.3ab | 564.3ab |
| 15 t ha⁻¹ | 7.0a | 7.4a | 7.9a | 1.2a | 1.1ab | 1.2a | 90.2a | 103.2a | 130.9a | 549.3a | 586.6a | 595.0a |
| LSD₀.₀₅ | 0.3 | | | 0.18 | | | 12.5 | | | 85 | | |

Values followed by the same letter in a column are not significantly (P < 0.05) different.

**Table 6. Soil NO₃-N, NH₄-N, PMN and P as influenced by compost application rates at the Mooketsi orchard from 2016 to 2018.**

| Treatment | NO₃-N | | | NH₄-N | | | PMN | | | P | | |
|---|---|---|---|---|---|---|---|---|---|---|---|---|
| | mg kg⁻¹ | | | mg kg⁻¹ | | | μgN gDM⁻¹ week⁻¹ | | | mg kg⁻¹ | | |
| | 2016 | 2017 | 2018 | 2016 | 2017 | 2018 | 2016 | 2017 | 2018 | 2016 | 2017 | 2018 |
| 0 t ha⁻¹ | 7.7b | 5.8b | 2.1c | 6.0b | 2.5b | 0.3c | 3.2c | 1.2c | 0.9c | 50.6c | 44.0d | 32.3d |
| 5 t ha⁻¹ | 9.4b | 11.7b | 14.0b | 10.9b | 18.6b | 38.2b | 9.4b | 11.6c | 19.6b | 61.0c | 177.6c | 198.0c |
| 10 t ha⁻¹ | 13.3ab | 19.4ab | 21.5ab | 43.2a | 51.5a | 60.2ab | 30.3a | 42.2a | 65.1a | 134.6b | 174.6b | 207.0b |
| 15 t ha⁻¹ | 19.7a | 22.1a | 28.9a | 59.4a | 60.7a | 75.8a | 40.6a | 56.3a | 72.3a | 193.0a | 189.3a | 225.0a |
| LSD₀.₀₅ | 9.2 | | | 30 | | | 18 | | | 22 | | |

Values followed by the same letter in a column are not significantly (P < 0.05) different. PMN = potentially mineralizable nitrogen.

study period. Significantly (p<0.05) higher quantities of the soil nutrient elements were recorded in the >5 t ha⁻¹ than <5 t ha⁻¹ compost application rates (Table 6). Soil NO₃-N, NH₄-N, PMN, and P were significantly (p<0.05) decreasing from 2016 to 2018 under the 0 t ha⁻¹ compost application rate (Table 6).

**Soil exchangeable cations (K, Ca, Mg and Na).** The K, Ca, Mg, and Na were significantly (p<0.05) varying in each year under the different compost application rates (Table 7). The K showed the smallest change of 5% increase in the compost applied plots in each year from 2016 to 2018 (Table 7). The Ca showed the greatest change of 25% increase per annum under the compost amended plots (Table 7).

**Soil micronutrients (Mn, Cu, Zn and Fe).** The Mn, Cu, Zn, and Fe content varied significantly (P<0.05) depending on the quantity of the eucalyptus wood-based compost applied

**Table 7. Soil exchangeable cations (K, Ca, Mg and Na) as influenced by compost application rates over a three year study (2016 to 2018) at the Mooketsi orchard.**

| Treatment | K | | | Ca | | | Mg | | | Na | | |
|---|---|---|---|---|---|---|---|---|---|---|---|---|
| | mg kg⁻¹ | | | | | | | | | | | |
| | 2016 | 2017 | 2018 | 2016 | 2017 | 2018 | 2016 | 2017 | 2018 | 2016 | 2017 | 2018 |
| 0 t ha⁻¹ | 252.0c | 162.0d | 124.3d | 306.7c | 226.7c | 121.3c | 302.0c | 212.6d | 182.6d | 26.6c | 17.6c | 10.3d |
| 5 t ha⁻¹ | 329.0b | 388.6c | 392.3c | 910.0b | 1124.0b | 1624.0b | 363.3b | 390.6c | 452.0c | 39.6b | 33.3b | 43.3c |
| 10 t ha⁻¹ | 366.3b | 481.3b | 498.3b | 1101.7a | 1131.0b | 1636.0a | 402.6ab | 496.0b | 517.3b | 45.6a | 65.6a | 70.6b |
| 15 t ha⁻¹ | 483.6a | 583.0a | 638.6a | 1125.0a | 1286.7a | 1808.3a | 469.6a | 597.6a | 648.3a | 48.6a | 73.0a | 92.6a |
| LSD₀.₀₅ | 51.3 | | | 142.4 | | | 50.3 | | | 8.2 | | |

Values followed by the same letter in a column are not significantly (P < 0.05) different.

**Table 8. Soil Mn, Cu, Zn and Fe content as influenced by compost application rates over a three year study (2016 to 2018) at the Mooketsi orchard.**

| Treatment | Mn | | | Cu | | | Zn | | | Fe | | |
|---|---|---|---|---|---|---|---|---|---|---|---|---|
| | mg kg$^{-1}$ | | | | | | | | | | | |
| | 2016 | 2017 | 2018 | 2016 | 2017 | 2018 | 2016 | 2017 | 2018 | 2016 | 2017 | 2018 |
| 0 t ha$^{-1}$ | 9.5d | 7.3d | 4.1d | 15.1c | 9.3c | 3.1d | 19.9c | 8.1d | 4.9d | 60.0d | 46.7d | 16.7d |
| 5 t ha$^{-1}$ | 27.6c | 46.3c | 56.7c | 19.0c | 33.5b | 42.8c | 22.0c | 39.0c | 49.8c | 75.6c | 84.5c | 100.2c |
| 10 t ha$^{-1}$ | 35.8b | 56.5b | 67.3b | 25.6b | 39.8b | 51.7b | 28.1b | 44.3b | 57.5b | 78.0b | 96.2b | 118.3b |
| 15 t ha$^{-1}$ | 43.5a | 66.1a | 79.4a | 34.9a | 42.9a | 62.6a | 36.7a | 54.6a | 67.5a | 86.1a | 99.8a | 128.7a |
| LSD$_{0.05}$ | 7.38 | | | 6.37 | | | 4.35 | | | 9.34 | | |

Values followed by the same letter in a column are not significantly (P < 0.05) different.

each year. The lowest and highest values of the micronutrients were observed in 2016 and 2018 respectively on the compost amended plots (Table 8).

## Politsi orchard

**Soil pH, EC, organic carbon and active carbon.** The soil pH was acidic (<7) throughout the 3 years under 0 t ha$^{-1}$ and 5 t ha$^{-1}$ application rates but became alkaline (>7) in 2017 and 2018 under 15 t ha$^{-1}$ compost application rate (Table 9). The EC was also the lowest (0.4 dS m$^{-1}$) and highest (1.3 dS m$^{-1}$) in 2018 on 0 t ha$^{-1}$ and 15 t ha$^{-1}$ compost applied plots respectively. The active C was increasing at an average annual rate of 3.4% and 10.2% on the 5 t ha$^{-1}$ and 15 t ha$^{-1}$ compost application rates respectively (Table 9).

**Soil NO$_3$-N, NH$_4$-N, PMN and P.** The eucalyptus wood-based compost significantly (P<0.05) increased soil NO$_3$-N, NH$_4$-N, PMN and P quantities in each year of the study (Table 10). Soil nitrates, ammonium nitrogen, and P were significantly (p<0.05) decreasing from 2016 to 2018 under no compost applied plots (Table 10).

**Soil exchangeable cations (K, Ca, Mg and Na).** Different application rates of the eucalyptus wood-based compost resulted in significantly (p<0.05) different quantities of K, Ca, Mg and Na in the soil (Table 11). The K was increasing at an annual average rate of a 3% increase in the compost applied plots from 2016 to 2018 (Table 11). Soil calcium had the greatest annual change rate of 25% during the 3-year study in the compost amended plots (Table 11).

**Soil micronutrients (Mn, Cu, Zn and Fe).** The Mn, Cu, Zn, and Fe soil content varied significantly (P<0.05) according to the amount of eucalyptus wood-based compost applied each year. Values of each soil micronutrient were lowest and highest in 2016 and 2018 respectively on the compost amended plots (Table 12).

**Table 9. Soil pH, EC, organic carbon and active carbon as influenced by compost application rates over three years (2016–2018) at the Politsi orchard.**

| Treatment | pH (H$_2$O) | | | EC (dS m$^{-1}$) | | | Organic C (mg kg$^{-1}$) | | | Active | | |
|---|---|---|---|---|---|---|---|---|---|---|---|---|
| C (mg kg$^{-1}$) | | | | | | | | | | | | |
| | 2016 | 2017 | 2018 | 2016 | 2017 | 2018 | 2016 | 2017 | 2018 | 2016 | 2017 | 2018 |
| 0 t ha$^{-1}$ | 5.5c | 5.6c | 5.4b | 0.5d | 0.5d | 0.4d | 2.8c | 1.3d | 0.7d | 129.6c | 92.3c | 70.3c |
| 5 t ha$^{-1}$ | 6.1b | 6.6b | 6.8b | 0.7c | 0.8c | 0.7c | 52.5b | 63.9c | 72.7c | 343.6b | 438.0b | 489.0b |
| 10 t ha$^{-1}$ | 6.6a | 6.8b | 7.0ab | 0.9b | 0.9b | 1.0b | 82.6a | 89.6b | 93.2b | 447.0a | 503.3ab | 560.0ab |
| 15 t ha$^{-1}$ | 6.9a | 7.1a | 7.6a | 1.1a | 1.2a | 1.3a | 91.4a | 100.4a | 139.2a | 542.7a | 579.0a | 653.6a |
| LSD$_{0.05}$ | 0.3 | | | 0.19 | | | 12.5 | | | 102 | | |

Values followed by the same letter in a column are not significantly (P < 0.05) different.

**Table 10. Soil NO₃-N, NH₄-N, PMN and P as influenced by compost application rates over a three year (2016–2018) study at the Politsi orchard.**

| Treatment | NO₃-N | | | NH₄-N | | | PMN | | | P | | |
|---|---|---|---|---|---|---|---|---|---|---|---|---|
| | mg kg⁻¹ | | | mg kg⁻¹ | | | μgN gDM⁻¹ week⁻¹ | | | mg kg⁻¹ | | |
| | 2016 | 2017 | 2018 | 2016 | 2017 | 2018 | 2016 | 2017 | 2018 | 2016 | 2017 | 2018 |
| 0 t ha⁻¹ | 6.7c | 5.9c | 3.1c | 7.0c | 3.5b | 0.6d | 3.0b | 1.1b | 0.9c | 49.9c | 40.6d | 31.7d |
| 5 t ha⁻¹ | 8.4c | 12.5c | 16.0b | 12.9c | 18.6b | 39.1c | 10.1b | 12.5b | 20.3b | 64.0c | 134.9c | 144.0c |
| 10 t ha⁻¹ | 16.3b | 18.9bc | 20.8b | 40.2b | 53.1a | 59.4b | 31.2a | 43.0a | 66.0a | 131.3b | 146.1b | 166.0b |
| 15 t ha⁻¹ | 20.3ab | 23.1ab | 30.1a | 58.1a | 63.8a | 78.2a | 40.1a | 57.3a | 73.3a | 183.0a | 192.0a | 220.2a |
| LSD₀.₀₅ | 8.2 | | | 13.1 | | | 17.6 | | | 36.3 | | |

Values followed by the same letter in a column are not significantly (P < 0.05) different. PMN = potentially mineralizable nitrogen.

**Table 11. Soil exchangeable cations (K, Ca, Mg and Na) as influenced by compost application rates over a three year study (2016 to 2018) at the Politsi orchard.**

| Treatment | K | | | Ca | | | Mg | | | Na | | |
|---|---|---|---|---|---|---|---|---|---|---|---|---|
| | mg kg⁻¹ | | | | | | | | | | | |
| | 2016 | 2017 | 2018 | 2016 | 2017 | 2018 | 2016 | 2017 | 2018 | 2016 | 2017 | 2018 |
| 0 t ha⁻¹ | 250.0c | 152.0d | 121.3d | 303.5c | 225.4c | 120.6c | 300.0c | 211.8d | 178.6d | 25.7c | 18.3c | 9.4d |
| 5 t ha⁻¹ | 324.0b | 389.6c | 396.3c | 904.0b | 1124.0b | 1604.0b | 360.8b | 388.7c | 443.2c | 38.9b | 30.6b | 41.6c |
| 10 t ha⁻¹ | 366.3b | 479.3b | 500.3b | 1100.4a | 1129.0b | 1630.0b | 412.1b | 486.8b | 514.4b | 46.4a | 68.3a | 72.8b |
| 15 t ha⁻¹ | 487.6a | 587.0a | 639.6a | 1121.6a | 1276.8a | 1800.1a | 465.3a | 595.8a | 644.6a | 49.5a | 78.0a | 90.3a |
| LSD₀.₀₅ | 48 | | | 140.9 | | | 49.1 | | | 8.6 | | |

Values followed by the same letter in a column are not significantly (P < 0.05) different.

Generally, the quantities of the measured soil chemical properties were significantly (P<0.05) increased with an increase in the quantity of the applied compost at both orchards. There were no significant differences observed on soil properties under the 0 t ha⁻¹ and 5 t ha⁻¹ application rates in 2016, but significant differences were noted in the subsequent years. The lowest and highest values of the soil chemical properties were recorded in 2016 and 2018 respectively under the different eucalyptus wood-based compost application rates at both the Mooketsi and Politsi orchards. The values of the soil chemical properties showed a constant increase with the increase in the quantity of the compost applied. The 0 t ha⁻¹ had the lowest values whereas the 15 t ha⁻¹ application rate recorded the highest values. In most cases, the soil

**Table 12. Soil micronutrients (Mn, Cu, Zn and Fe) as influenced by compost application rates over a three year study (2016–2018) at the Politsi orchard.**

| Treatment | Mn | | | Cu | | | Zn | | | Fe | | |
|---|---|---|---|---|---|---|---|---|---|---|---|---|
| | Femg kg⁻¹ | | | | | | | | | | | |
| | 2016 | 2017 | 2018 | 2016 | 2017 | 2018 | 2016 | 2017 | 2018 | 2016 | 2017 | 2018 |
| 0 t ha⁻¹ | 9.2d | 8.3d | 3.9d | 14.8c | 9.0c | 3.7d | 19.9c | 9.1d | 3.9d | 59.6d | 45.6d | 18.6c |
| 5 t ha⁻¹ | 25.6c | 40.9c | 53.8c | 16.9c | 34.2b | 41.4c | 20.8c | 37.0c | 46.3c | 74.4c | 80.9c | 99.8b |
| 10 t ha⁻¹ | 37.8b | 54.3b | 64.8b | 25.0b | 37.6b | 49.1b | 28.4b | 40.2b | 55.4b | 78.6b | 98.1b | 120.4a |
| 15 t ha⁻¹ | 40.1a | 62.1a | 74.2a | 34.3a | 40.1a | 62.1a | 34.7a | 54.7a | 67.4a | 84.9a | 100.5a | 129.5a |
| LSD₀.₀₅ | 7.30 | | | 6.00 | | | 4.12 | | | 9.38 | | |

Values followed by the same letter in a column are not significantly (P < 0.05) different.

properties were significantly the same under the 10 t ha$^{-1}$ and 15 t ha$^{-1}$ compost application rates in 2017 and 2018.

## Discussion

### Soil pH, EC, organic carbon and active carbon

The addition of compost significantly raised soil pH and the application rate of 15 t ha$^{-1}$ recorded the highest pH increase at the Mooketsi and Politsi orchards. These observations are contrary to [23] who noted a decrease in soil pH with an increase of the organic matter applied. In this study, the pH could have been affected by the quality of the applied compost. The eucalyptus wood-based compost was acidic (6.91) so larger (>5 t ha$^{-1}$) quantities were acidifying the soil by releasing the H$^{+}$ ions in each year of the application causing the noted increase of soil pH at 15 t ha$^{-1}$ compost application rate. The results agree with [24] who reported an increase in soil pH value even at moderate compost applications.

Soil EC measures directly the salinity and indirectly the soluble salts within the soil. In this study, the application of the compost increased the EC of the soil. An EC> 1 dS m$^{-1}$ indicates saline soil. The application of the eucalyptus wood-based compost at 15 t ha$^{-1}$ therefore resulted to saline soils at both the Mooketsi and Politsi orchards (Tables 5 and 9). The increase in soil EC in compost amended plots could be due to high salinity within the compost. The results are similar to [25] who noted a proportional increase in soil EC with the quantities of compost applied. The rise in soil EC level was typical because saline compost was used in the study [25]. Soil organic carbon (SOC) and active carbon (AC) were significantly (p<0.05) increasing in proportion to the quantity of the applied eucalyptus wood-based compost throughout the study period at both orchards. Active carbon measures the portion of organic matter which will function as an easily available source of soil microbes [26, 27]. The observed increase in SOC and AC could be due to the cumulative buildup from the added compost in each year of the study. [28], and [29] also observed an increase in SOC and AC as a result of compost application. [9] also reported on the rise of organic carbon following the applying of manure under different land-use systems.

**Soil nitrate-N (NO$_3$-N), ammonium-N (NH$_4$-N), potentially mineralizable nitrogen (PMN) and P.** The increase of the soil nitrate, NH$_4$-N, and PMN was directly proportional to the applied quantities of the eucalyptus wood-based compost. The eucalyptus wood-based compost contained a high concentration (>25 ppm) of NO$_3$-N and this could have resulted in an increase in O$_3$-N, NH$_4$-N, and PMN concentrations in the soil in each year of application. Similar results were reported by [30] when noted increase in soil nitrates with increased applied quantities of N enriched compost. Potentially mineralizable nitrogen was increasing each year in direct proportion to the quantity of the eucalyptus wood-based compost applied. The PMN was highest under 15 t ha$^{-1}$ during the 2018 season at both orchards. This suggests that the carbon-nitrogen (C: N) of the compost (C: N = 23.6:1) was ideal and that promoted a high rate of nitrogen mineralization. Organic matter with C: N ratios of 20:1 to 30:1 were observed to decompose quickly and completely, and release considerable nitrogen through mineralization. However, the application of such compost did not result in an increase in the soil organic matter concentration and this could explain the observed slow increase in the organic carbon and active carbon of the soil (Tables 5 and 9). Carbon-nitrogen the ratio of the applied compost to the soil was important because it influenced the rate at which the compost decomposed and the amount of nitrogen recycled from the compost.

The addition of eucalyptus wood-based compost caused a small increase in soil P levels from 2016 to 2018 at the Mooketsi and Politsi orchards (Tables 6 and 10). Organic

supplements have been reported to increase P availability in P-fixing soils [31] and humic substances were observed to enhance the bioavailability of P fertilizers in acidic soils [32]. Our results are suggesting that applying the different rates of eucalyptus wood-based compost decreased the P adsorption of soil in all compost amended plots. The soils at Mooketsi and Politsi are Nitisols and Ferralsols respectively which are typically acidic soils (Table 2). The soil types are rich in iron and aluminum oxides, which have high degrees of crystallinity and high P-adsorption capacities. However, the application of the eucalyptus wood-based compost ($\geq 5$ t ha$^{-1}$) could have reduced the P-adsorption capacity of these soils. The possible explanation for this phenomenon could be that the combination of compost with iron and aluminum oxides in the acidic soils might reduce the P-adsorption capacities of the oxides. Similarly, [25] noted a rise in soil P with an increase of the compost applied. [32] also reported an increase of soil P concentration in a direct proportion to the husk compost application rates.

## Soil exchangeable cations (K, Ca, Mg and Na)

There was a direct proportion increase between the eucalyptus wood-based compost application rates and K, Ca, Mg and Na in each year at both Mooketsi and Polisti orchards (Tables 7 and 11). The increase of the soil exchangeable cations especially at high (15 t ha$^{-1}$) eucalyptus wood-based compost application could be a result of the high concentration of the cations in compost. The results are in agreement with [33] who reported the high concentration levels of exchangeable K, Ca, and Mg in soils that were amended with different rates of farmyard manure-based compost annually. The increase in the soil exchangeable cations levels e.g Ca in eucalyptus wood-based compost amended plots can be due to the discharge of organic acids and $CO_2$ from the compost in the successive years (2016–2018), hence raising the pH of the encompassing soil which ends up in solubilizing Ca and Mg from carbonates. Our results are similar to [34] who reported an increase in soil Mg and Ca with increasing quantities of compost applied.

## Soil micronutrients (Mn, Cu, Zn, and Fe)

The concentration of the soil micronutrients (Mn, Cu, Zn, and Fe) followed a similar trend to the soil exchangeable cations at the Mooketsi and Politsi (Tables 8 and 12). Soil micronutrient levels were increasing in proportion to the applied quantities of the eucalyptus throughout the study period. The highest concentrations of the Mn, Cu, Zn, and Fe were recorded at 15 t ha$^{-1}$ during 2018 at both orchards. The increase in soil Mn level suggests the dissolution of Mn precipitates as a result of microbial activity that changes gaseous composition that can occur in organic matter-rich soils. However, results from [35] suggested that the Mn can decrease with an increase in organic applied. The disagreement of these results cannot be ascertained in this study but probability it could due to differences in compost qualities. The eucalyptus wood-based compost used in this study enhanced soil microbial activities (low C: N) (Table 2). The amount of soil organic matter is one of the foremost important factors that influence the mobility of Cu, Zn, and Fe [35, 36]. The increase in Cu, Zn, and Fe levels with the increase in the eucalyptus wood based compost application rates could be explained by the high the cation exchange capacity of the compost and its ability to make chelates with Cu, Zn, and Fe. A similar trend was observed by [37] while investigating different levels of sewage sludge.

## Conclusion

The effects of eucalyptus wood-based compost on selected chemical properties of soils were varying in proportion with the application rates. Values of the selected soil chemical properties

were highest and lowest at 15 t ha$^{-1}$ and 0 t ha$^{-1}$ respectively. The effect of compost application rates on the soil chemical properties was consistent at both study sites and there were no significant (p>0.05) differences between 10 t ha$^{-1}$ and 15 t ha$^{-1}$ application rates. Therefore applying compost at least 10 t ha$^{-1}$ is an idea in improving the soil chemical properties at the Mooketsi and Politsi orchards. However, there are other factors that affect organic matter decomposition e.g soil temperature and microbial activity which were not quantified in this study hence need for further studies considering such factors.

## Acknowledgments

The authors are grateful to the Zimbabwe Open University for providing an updated statistical software during data analysis.

## Author Contributions

**Conceptualization:** A. Manyevere.

**Data curation:** P. M. Mohale.

**Formal analysis:** C. Parwada.

**Investigation:** M. G. Zerizghy.

**Methodology:** A. Manyevere, C. Parwada, M. G. Zerizghy.

**Project administration:** P. M. Mohale.

**Software:** C. Parwada.

**Supervision:** M. G. Zerizghy.

**Writing – review & editing:** C. Parwada.

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
