## [Decision Letter · Decision Letter 0]

3 Jan 2022

PONE-D-21-30996Effect of Eucalyptus wood-based compost application rates on soil chemical properties in Mooketsi and Politsi avocado orchards, Limpopo province, South AfricaPLOS ONE

Dear Dr. Manyevere,

Thank you for submitting your manuscript to PLOS ONE. After careful consideration, we feel that it has merit but does not fully meet PLOS ONE’s publication criteria as it currently stands. Therefore, we invite you to submit a revised version of the manuscript that addresses the points raised during the review process.

We look forward to receiving your revised manuscript.

Kind regards,

Muhammad Riaz

Academic Editor

PLOS ONE

https://journals.plos.org/plosone/s/file?id=ba62/PLOSOne_formatting_sample_title_authors_affiliations.pdf"

4. Thank you for stating the following in the Funding Section of your manuscript:

“This research was funded by National Research Fund (NRF) and ZZ2 Bertie van Zyl (Pty) Ltd, South Africa.”

“he funders had no role in study design, data collection and analysis, decision to publish, or preparation of the manuscript”

6. Please include a copy of Table 7.

Additional Editor Comments:

Our reviewer(s) have commented on your manuscript and they recommend that you revise your manuscript. I invited you to submit your revise manuscript by giving due consideration to comments and suggestions from reviewers. I also urge you to check the manuscript for English language mistakes.

Reviewers' comments:

Reviewer's Responses to Questions

**Comments to the Author**

1. Is the manuscript technically sound, and do the data support the conclusions?

Reviewer #1: Yes

2. Has the statistical analysis been performed appropriately and rigorously? 

Reviewer #1: Yes

3. Have the authors made all data underlying the findings in their manuscript fully available?

Reviewer #1: Yes

4. Is the manuscript presented in an intelligible fashion and written in standard English?

Reviewer #1: Yes

5. Review Comments to the Author

Reviewer #1: This study showed how eucalyptus wood based compost affect on soil chemical properties in two different avocado orchards. All application rates of this compost generally increased this properties in soil. Authors suggested that 10 t application can be optimal for improving the soil chemical properties. After these corrections and suggestions, this manuscript can be published.

Line 114-115: Is there any field capacity data? If there is, can you add it?

Line 196: under t ha ? which application?

Line 310: Additions of articles titled as "The First Effect of Eucalyptus camaldulensis Leaves and Eucalyptol Additions on Soil Carbon Mineralization" and "Short-Term Eucalyptus and Phragmites Biochar’s Efficiency in Mineralization of Soil Carbon" can help to explain how eucalyptus compost affect on soil organic carbon accumulation in soil.

Due to the high salinity of compost, addition of this compost into soil may have directly affect the activity of soil microorganisms in this study. Soil biological properties are directly linked with soil chemical properties. Is there any suggestion for reducing the salinity of this compost while it was produced? In addition to that, is there any data about the EC of irrigation water in this study? Is it possible that salinity of irrigation water may have increased soil salinity?

All application rates of this compost significantly increased soil chemical properties compared to control. However, EC can be a big problem in plant production. Based on EC data in this study, I think 5 t/ha application rate of eucalyptus compost is more optimum dose for improving soil chemical properties rather than higher doses.

6. PLOS authors have the option to publish the peer review history of their article (what does this mean?). If published, this will include your full peer review and any attached files.

Reviewer #1: No

---

## [Author Response · Author response to Decision Letter 0]

20 Feb 2022

Please ensure that your manuscript meets PLOS ONE's style requirements, including those for file naming. The PLOS ONE’s style was adopted in preparing the manuscript

In your Methods section, please provide additional information regarding the permits you obtained for the work. Please ensure you have included the full name of the authority that approved the field site access and, if no permits were required, a brief statement explaining why. The additional information was added as was suggested

We note that the grant information you provided in the ‘Funding Information’ and ‘Financial Disclosure’ sections do not match.

When you resubmit, please ensure that you provide the correct grant numbers for the awards you received for your study in the ‘Funding Information’ section. No specific grant was given for this study

Thank you for stating the following in the Funding Section of your manuscript:

“This research was funded by National Research Fund (NRF) and ZZ2 Bertie van Zyl (Pty) Ltd, South Africa.”

“he funders had no role in study design, data collection and analysis, decision to publish, or preparation of the manuscript”

 The research did not receive any specific grant although the first author was supported by the NRF and the ZZ2 Bertie van Zyl (Pty) Ltd, South Africa but not under any specific grant as was stated

Please include your full ethics statement in the ‘Methods’ section of your manuscript file. In your statement, please include the full name of the IRB or ethics committee who approved or waived your study, as well as whether or not you obtained informed written or verbal consent. If consent was waived for your study, please include this information in your statement as well. The ethics statement was written in the Methods section as was indicated

Please include a copy of Table 7. Table 7 was inserted as suggested

Rviewer #1 

Line 114-115: Is there any field capacity data? If there is, can you add it? The field capacity information was included as suggested. See the yellow colour in the attached manuscript

Line 196: under t ha ? which application? The missing tonnage was added- see the correction in yellow

Line 310: Additions of articles titled as "The First Effect of Eucalyptus camaldulensis Leaves and Eucalyptol Additions on Soil Carbon Mineralization" and "Short-Term Eucalyptus and Phragmites Biochar’s Efficiency in Mineralization of Soil Carbon" can help to explain how eucalyptus compost affect on soil organic carbon accumulation in soil. The additional literature was added as was suggested. See yellow marks in the attached manuscript. The given sources were consulted.

Due to the high salinity of compost, addition of this compost into soil may have directly affect the activity of soil microorganisms in this study. Soil biological properties are directly linked with soil chemical properties. Is there any suggestion for reducing the salinity of this compost while it was produced? In addition to that, is there any data about the EC of irrigation water in this study? Is it possible that salinity of irrigation water may have increased soil salinity?

All application rates of this compost significantly increased soil chemical properties compared to control. However, EC can be a big problem in plant production. Based on EC data in this study, I think 5 t/ha application rate of eucalyptus compost is more optimum dose for improving soil chemical properties rather than higher doses. Yes, we agree with the reviewer’s suggestion but the EC of the soils were < 1 dS/m (the maximum salinity level where salinity problem may occur in plant production) in 0. 5 and 10 t/ha eucalyptus-wood-based compost (See table 5 and 9). Hence application rates up to 15 t/ha may not be such problematic salinity according to our findings.

The EC of the irrigation water was added as was suggested.

---

## [Decision Letter · Decision Letter 1]

8 Mar 2022

Effect of Eucalyptus wood-based compost application rates on soil chemical properties in Mooketsi and Politsi avocado orchards, Limpopo province, South Africa

PONE-D-21-30996R1

Dear Dr. Manyevere,

We’re pleased to inform you that your manuscript has been judged scientifically suitable for publication and will be formally accepted for publication once it meets all outstanding technical requirements.

Kind regards,

Muhammad Riaz

Academic Editor

PLOS ONE

Additional Editor Comments (optional):

I am pleased to inform you that your revised manuscript has been accepted for publication.

Reviewers' comments:

Reviewer's Responses to Questions

**Comments to the Author**

1. If the authors have adequately addressed your comments raised in a previous round of review and you feel that this manuscript is now acceptable for publication, you may indicate that here to bypass the “Comments to the Author” section, enter your conflict of interest statement in the “Confidential to Editor” section, and submit your "Accept" recommendation.

Reviewer #1: All comments have been addressed

2. Is the manuscript technically sound, and do the data support the conclusions?

Reviewer #1: Yes

3. Has the statistical analysis been performed appropriately and rigorously? 

Reviewer #1: Yes

4. Have the authors made all data underlying the findings in their manuscript fully available?

Reviewer #1: Yes

5. Is the manuscript presented in an intelligible fashion and written in standard English?

Reviewer #1: Yes

6. Review Comments to the Author

Reviewer #1: (No Response)

7. PLOS authors have the option to publish the peer review history of their article (what does this mean?). If published, this will include your full peer review and any attached files.

Reviewer #1: No

---

## [Editor Report · Acceptance letter]

17 Mar 2022

PONE-D-21-30996R1 

Effect of Eucalyptus wood-based compost application rates on soil chemical properties
in semi-organic Avocado plantations, Limpopo province, South Africa 

Dear Dr. Manyevere:

I'm pleased to inform you that your manuscript has been deemed suitable for publication in PLOS ONE. Congratulations! Your manuscript is now with our production department. 

Kind regards, 

on behalf of

Dr. Muhammad Riaz 

Academic Editor

PLOS ONE